# CONVERSATIONAL ORIENTATION REASONING: EGOCENTRIC-TO-ALLOCENTRIC NAVIGATION WITH MULTIMODAL CHAIN-OF-THOUGHT

## ABSTRACT

Conversational agents must translate egocentric utterances (e.g., "on my right") into allocentric orientations (N/E/S/W). This challenge is particularly critical in indoor or complex facilities where GPS signals are weak and detailed maps are unavailable. While chain-of-thought (CoT) prompting has advanced reasoning in language and vision tasks, its application to multimodal spatial orientation remains underexplored. We introduce **Conversational Orientation Reasoning (COR)**, a new benchmark designed for Traditional Chinese conversational navigation projected from real-world environments, addressing egocentric-to-allocentric reasoning in non-English and ASR-transcribed scenarios. We propose a multimodal chain-of-thought (MCoT) framework, which integrates ASR-transcribed speech with landmark coordinates through a structured three-step reasoning process: (1) extracting spatial relations, (2) mapping coordinates to absolute directions, and (3) inferring user orientation. A curriculum learning strategy progressively builds these capabilities on Taiwan-LLM-13B-v2.0-Chat, a mid-sized model representative of resource-constrained settings. Experiments show that MCoT achieves 100% orientation accuracy on clean transcripts and 98.1% with ASR transcripts, substantially outperforming unimodal and non-structured baselines. Moreover, MCoT demonstrates robustness under noisy conversational conditions, including ASR recognition errors and multilingual code-switching. The model also maintains high accuracy in cross-domain evaluation and resilience to linguistic variation, domain shift, and referential ambiguity. These findings highlight the potential of structured MCoT spatial reasoning as a path toward interpretable and resource-efficient embodied navigation.

## 1 INTRODUCTION

Humans naturally describe spatial environments in *egocentric* (agent-centric) terms (e.g., "The exit is on my right"), whereas navigation systems typically operate on *allocentric* (world-centric) orientations such as north, south, east, and west. Conversational navigation has emerged as a promising paradigm that enables users to specify goals through dialogue, offering a natural and human-centered means of guidance in unfamiliar environments Sundar et al. (2024); Sheshadri & Hara (2024); Kaniwa et al. (2024); Liu et al. (2024); Levi & Kadar (2025). However, the crucial problem of grounding egocentric language into allocentric orientation remains underexplored. Current approaches typically assume access to GPS, detailed maps, or fixed global frames de Vries et al. (2018); Chen et al. (2020), and have concentrated primarily on English-based scenarios. Recent progress has also relied heavily on large-scale models Ghosh et al. (2024); Tang et al. (2023), which show strong reasoning abilities but demand substantial computational resources, hindering deployment in resource-constrained settings such as mobile navigation and edge devices.

Research in embodied AI and MCoT has advanced vision-language navigation and action planning Mu et al. (2023); Sun et al. (2024); Liu et al. (2025); Shen et al. (2025); Pareek et al. (2024), but orientation reasoning from natural language has been largely overlooked. These approaches typically assume that the agent's orientation is already known or operate on high-level action spaces rather than inferring fundamental spatial relationships de Vries et al. (2018); Chen et al. (2020). Meanwhile, large audio-language models (LALMs) Zhang et al. (2023); Xie & Wu (2024); Fu et al.

(2025); Défossez et al. (2024) have advanced speech understanding and dialogue Tang et al. (2024); Gong et al. (2023); Ghosh et al. (2024); Kong et al. (2024), yet their reasoning abilities remain limited to perception-level tasks such as transcription or summarization yu Huang et al. (2024); Yang et al. (2024); Wang et al. (2025); Shi et al. (2025). While recent efforts like Audio-CoT Ma et al. (2025) show promise for enhanced speech-based reasoning, the challenge of transforming egocentric spatial descriptions into allocentric orientation inference remains unaddressed.

To address this gap, we introduce **Conversational Orientation Reasoning (COR)**, a new benchmark for egocentric-to-allocentric orientation reasoning in Traditional Chinese conversational navigation. COR is derived from real-world urban transportation environments in Taiwan, projected into structured grid representations. Unlike prior studies that rely on vision or raw audio, COR combines ASR-transcribed egocentric language with structured landmark coordinates, evaluating both clean text and ASR transcripts to simulate realistic recognition errors. COR addresses the lack of non-English benchmarks in multimodal spatial reasoning, particularly under noisy ASR conditions.

Our study is guided by three research questions:

- **RQ1 (Effectiveness):** How effective is multimodal CoT prompting for orientation reasoning compared to unimodal and unstructured baselines?
- **RQ2 (Component analysis):** What are the contributions of ASR preprocessing, multimodal fusion, and structured CoT steps?
- **RQ3 (Robustness and generalization):** How robust is the approach to linguistic variation, and how well does it generalize across different spatial domains?

Our contributions are as follows:

1. **Task and benchmark.** We introduce the COR benchmark for egocentric-to-allocentric orientation reasoning, combining ASR-transcribed speech with landmark coordinates.
2. **Framework.** We develop a multimodal CoT framework with a structured three-step reasoning process that integrates noisy transcripts with spatial signals for orientation inference.
3. **Evaluation.** We provide extensive experiments in Traditional Chinese, demonstrating effectiveness, component contributions, and robustness validation across linguistic variation, cross-domain generalization, and referential ambiguity beyond English-centric research.

## 2 RELATED WORK

**Navigation and Spatial Orientation.** Natural language navigation tasks have long driven progress in embodied AI. The Room-to-Room (R2R) benchmark Anderson et al. (2018) first established visually grounded navigation instructions in real indoor environments, while Cooperative Vision-and-Dialog Navigation (CVDN) Thomason et al. (2019) extended this to conversational settings where agents interpret multi-turn human dialogues. Talk the Walk de Vries et al. (2018) grounds tourist utterances with masked attention but assumes orientation-agnostic actions. Touchdown Chen et al. (2020) extends navigation to urban environments with graph-based orientation, while SpatialRGPT Cheng et al. (2024) leverages 3D scene graphs for spatial grounding. Speaker-Follower models Fried et al. (2018) improve instruction following by embedding panoramic orientation, and Ego4D Grauman et al. (2022) provides large-scale egocentric video datasets for studying orientation and attention. Despite these advances, existing frameworks assume known agent orientations or operate on high-level action spaces such as Left, Right, Up, and Down rather than addressing the fundamental egocentric-to-allocentric transformation challenge de Vries et al. (2018); Fried et al. (2018). In contrast, our work tackles the egocentric-to-allocentric challenge, using MCoT to infer absolute directions from noisy ASR transcripts and landmark coordinates.

**Chain-of-Thought and Multimodal Reasoning.** CoT prompting has proven effective for enhancing reasoning in large language models (LLMs). Increasing the number of reasoning steps in demonstrations improves accuracy across diverse benchmarks Jin et al. (2024), and subsequent studies show that performance critically depends on the logical consistency and contextual relevance of rationales Wang et al. (2023); Prystawski et al. (2023); Tang et al. (2023). Recent extensions to MCoT enable embodied agents to reason jointly over multiple modalities. For instance, EmbodiedGPT Mu

et al. (2023) and E-CoT Lin et al. (2024) segment tasks into subgoals, while Emma-X Sun et al. (2024) incorporates grounded planning and predictive movement. SpatialCoT Liu et al. (2025) focuses on coordinate alignment, and MCoCoNav Shen et al. (2025) integrates semantic maps for multi-robot collaboration. Together, these advances highlight the growing importance of MCoT for embodied reasoning. However, none of these approaches tackles the transformation from egocentric natural language descriptions to allocentric orientation inference under noisy ASR conditions.

**Large Audio-Language Models.** Large audio-language models (LALMs) extend this line of research by incorporating speech as an additional input. They have advanced transcription, classification, and interactive dialogue Zhang et al. (2023); Xie & Wu (2024); Fu et al. (2025); Défossez et al. (2024), showing strong performance on perception-level tasks such as speech recognition and summarization Tang et al. (2024); Gong et al. (2023); Ghosh et al. (2024); Kong et al. (2024); yu Huang et al. (2024); Yang et al. (2024); Wang et al. (2025); Shi et al. (2025). Audio-CoT Ma et al. (2025) suggests that CoT can aid speech-derived reasoning, but the improvements remain modest and fail to generalize to complex reasoning tasks. While existing LALMs excel at perception, they lack mechanisms for structured multi-step reasoning, limiting their effectiveness in noisy or ambiguous conditions. Our work addresses this by treating ASR transcripts as a noisy textual modality, integrating them with landmark coordinates, and applying MCoT to enable orientation inference.

## 3 METHOD

### 3.1 OVERVIEW

We propose a MCoT framework for egocentric-to-allocentric orientation reasoning in conversational navigation. To simulate realistic speech-driven conditions, we synthesize speech from clean descriptions and transcribe it using automatic speech recognition (ASR), thereby introducing natural transcription errors. This controlled approach allows us to systematically evaluate performance under varying levels of ASR noise while maintaining reproducible experimental conditions. The resulting transcript $A'$ is combined with structured spatial coordinates $T$ that describe the user's position and nearby landmarks. The model then generates both an interpretable reasoning trace and the final allocentric orientation prediction $D^* \in \{\text{north, east, south, west}\}$.

As illustrated in Figure 1, the framework consists of three modules: (1) *Speech synthesis and transcription*, where original descriptions are converted into transcripts via text-to-speech (TTS) and ASR; (2) *Multimodal input preparation*, which encodes transcripts and spatial coordinates into a unified representation; and (3) *Orientation reasoning*, where the model applies structured CoT inference to derive the final allocentric prediction. Unlike prior LALMs that emphasize transcription quality, our framework focuses on spatial reasoning under noisy transcripts.

### 3.2 TASK

Let $A$ denote the clean egocentric description, $A'$ the ASR transcript, $T$ the spatial coordinate input, and $D^*$ the ground-truth allocentric orientation. We formalize orientation reasoning on a discrete grid derived from the Gongguan MRT area, a busy transportation hub in Taiwan (Figure 2).

**Environment.** The real-world area is projected into a $10 \times 10$ grid $\mathcal{G}$, with user position $u = (x_u, y_u) \in \mathcal{G}$ and landmarks $\mathcal{L} = \{\ell_i\}$ with coordinates $p(\ell_i) \in \mathcal{G}$.

**Egocentric description.** The user describes their context with an egocentric relation $q \in \{\text{FRONT, BACK, LEFT, RIGHT}\}$ and a reference landmark $\ell_r$. After TTS and ASR processing, we obtain transcript $A'$. For example: *"I am at Exit 2, and restaurant 5 is on my right"* $\rightarrow D^* = \text{north}$, which implies that the user is facing north.

**Mapping rule.** We first compute the relative vector between the landmark and the user,

$$\Delta = p(\ell_r) - u = (\Delta_x, \Delta_y).$$

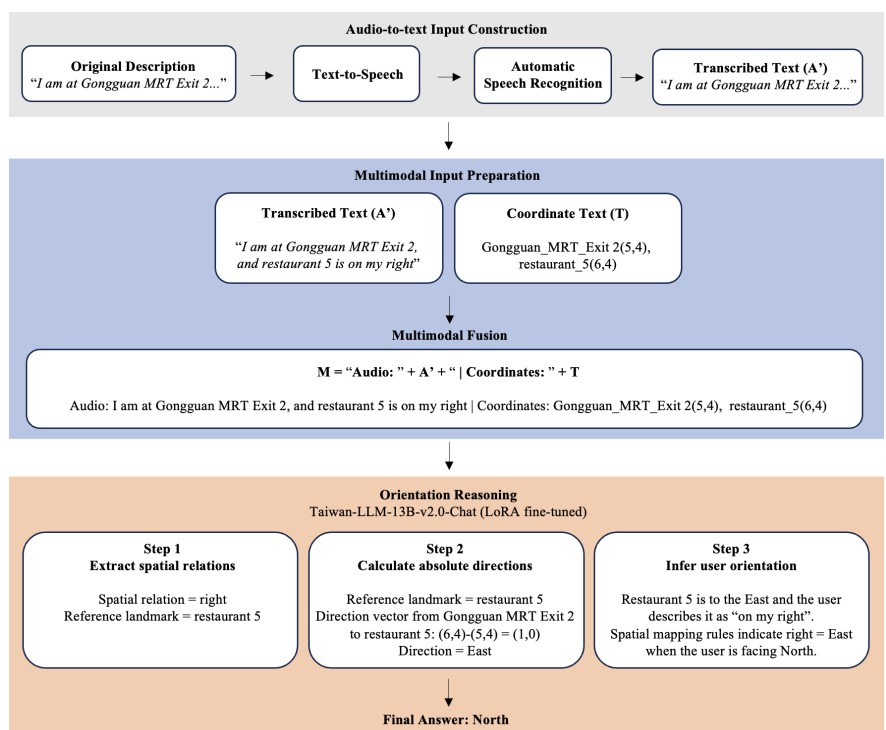

Figure 1: Pipeline of our MCoT framework. It consists of three modules: (1) speech synthesis and transcription, (2) multimodal input preparation and fusion, and (3) orientation reasoning.

The absolute direction of $\ell_r$ is then defined as

$$\text{AbsDir}(\Delta) = \begin{cases} E & \text{if } |\Delta_x| > |\Delta_y| \ \wedge \ \Delta_x > 0, \\ W & \text{if } |\Delta_x| > |\Delta_y| \ \wedge \ \Delta_x < 0, \\ N & \text{if } |\Delta_y| > |\Delta_x| \ \wedge \ \Delta_y > 0, \\ S & \text{if } |\Delta_y| > |\Delta_x| \ \wedge \ \Delta_y < 0, \end{cases}$$

and the user's orientation $D^*$ is derived by rotating $d_{\text{abs}} = \text{AbsDir}(\Delta)$ according to relation $q$:

$$D^* = \begin{cases} d_{\text{abs}} & q = \text{FRONT}, \\ \text{Rot}(d_{\text{abs}}, 180°) & q = \text{BACK}, \\ \text{Rot}(d_{\text{abs}}, +90°) & q = \text{LEFT}, \\ \text{Rot}(d_{\text{abs}}, -90°) & q = \text{RIGHT}. \end{cases}$$

Since the environment is represented as a discrete $10 \times 10$ grid with only four cardinal neighbors $(0, \pm 1), (\pm 1, 0)$, i.e., a Manhattan grid, landmarks are always axis-aligned and diagonal cases do not occur. All ground-truth orientations D* are automatically derived from these mapping rules. Each instance includes a step-by-step reasoning trace used to supervise CoT generation. To ensure quality, all automatically generated instances are verified by human annotators for correctness.

Table 1 lists the full set of relative-to-absolute mapping rules.

### 3.3 TRAINING

In preliminary experiments, directly training the model end-to-end led to unstable learning and poor generalization. We therefore adopt curriculum learning with stage-wise fine-tuning on Taiwan-LLM-13B-v2.0-Chat to incrementally build orientation reasoning capabilities. All training stages use clean text, while noisy ASR transcripts are introduced only during evaluation. This design choice allows the model to first master fundamental spatial reasoning without transcription noise, then tests performance when ASR errors are introduced.

| Facing | Front | Back | Right | Left |
|--------|-------|------|-------|------|
| North  | N     | S    | E     | W    |
| East   | E     | W    | S     | N    |
| South  | S     | N    | W     | E    |
| West   | W     | E    | N     | S    |

Table 1: Relative-to-absolute direction mapping rules.

Figure 2: Task environment. Gongguan MRT area projected into a $10 \times 10$ grid map for testing.

**Stage-wise fine-tuning.** Let $f_{\theta^{(i)}}$ denote the model at stage $i$.

**(S1) Relation extraction.** At the first stage, the model identifies the egocentric relation and reference landmark from clean description A:

$$r_1 = f_{\theta^{(0)}}(A) \rightarrow (q, \ell_r). \tag{1}$$

**(S2) Coordinate mapping.** At the second stage, the model converts user and landmark coordinates into an absolute direction:

$$r_2 = f_{\theta^{(1)}}(u, p(\ell_r)) \rightarrow d_{\text{abs}} \in \{\text{N,E,S,W}\}. \tag{2}$$

**(S3) Orientation reasoning.** At the third stage, the model predicts the final orientation given the absolute direction and egocentric relation:

$$r_3 = f_{\theta^{(2)}}(d_{\text{abs}}, q) \rightarrow D^*. \tag{3}$$

**(S4) End-to-end integration.** Finally, transcripts and coordinates are serialized into a single textual multimodal input:

$$M = \text{``Audio: ''} + A' + \text{`` | Coordinates: ''} + T \tag{4}$$

where "+" denotes string concatenation and $T$ serializes $(u, \{\ell_i, p(\ell_i)\})$ into tokens. The model then produces a full reasoning trace:

$$S = f_{\theta^{(3)}}(M), \tag{5}$$

which includes $(r_1, r_2, r_3, D^*)$.

| Standard Prompting | Multimodal Chain-of-Thought Prompting |
|---|---|
| **Model Input** | **Model Input** |
| Audio: "*I'm at Gongguan MRT Exit 2, and restaurant 5 is on my right.*" 
 Coordinates: Gongguan_MRT_Exit 2(5,4), restaurant_5(6,4) 
 Query: Which direction is the user facing? | Audio: "*I'm at Gongguan MRT Exit 2, and restaurant 5 is on my right.*" 
 Coordinates: Gongguan_MRT_Exit 2(5,4), restaurant_5(6,4) 
 Query: Which direction is the user facing? Please reason step by step: |
| **Model Output** | **Model Output** |
| Final answer: East ✗ | **Step 1: Extract spatial relations** 
 Spatial relation = right 
 Reference landmark = restaurant 5 
 **Step 2: Calculate absolute directions** 
 Reference landmark = restaurant 5 
 Direction vector from Gongguan MRT Exit 2 to restaurant 5: (6,4)-(5,4) = (1,0) 
 Direction = East 
 **Step 3: Infer user orientation** 
 Restaurant 5 is to the East and the user describes it as "on my right". 
 Spatial mapping rules indicate right = East when the user is facing North. 
 Therefore, the user is facing North. 

 Final answer: North ✓ |

Figure 3: Comparison of standard prompting and MCoT. Standard prompting fails under ambiguous egocentric descriptions, whereas MCoT uses structured steps for better accuracy and interpretability.

**Objective.** Training supervision is applied to intermediate reasoning steps and final predictions:

$$\mathcal{L} = -\sum_{t=1}^{|S|} \log P(s_t \mid s_{<t}, M). \tag{6}$$

### 3.4 MULTIMODAL CHAIN-OF-THOUGHT

CoT prompting enhances reasoning by decomposing tasks into interpretable steps Jin et al. (2024); Wu et al. (2023). Extending this to multimodal settings improves interpretability and stability Xie et al. (2025); Cui et al. (2024). Building on these insights, our MCoT framework handles ASR noise, provides interpretable rationales, and localizes errors to specific reasoning stages. This makes MCoT well suited for conversational navigation in indoor or GPS-limited environments.

**Three-step reasoning.** Our MCoT decomposes orientation reasoning into three steps: (1) Relation extraction: identify relation $q$ and landmark $\ell_r$ from input transcript, (2) Coordinate mapping: compute $d_{\text{abs}}$ from $(u, p(\ell_r))$, and (3) Orientation reasoning: infer $D^*$ given $(d_{\text{abs}}, q)$.

**Comparison to standard prompting.** As shown in Figure 3, standard prompting directly maps input to output, often failing under ambiguous egocentric language. In contrast, MCoT introduces intermediate steps that align spatial relations with mapping rules, yielding more accurate and interpretable predictions. While our experiments focus on a $10 \times 10$ grid for tractability, the framework can be extended to larger or continuous maps by adapting the coordinate mapping rules.

## 4 EXPERIMENTAL RESULTS

### 4.1 DATASET

We evaluate on the COR benchmark, which we construct for egocentric-to-allocentric orientation reasoning in conversational navigation. COR contains 4,600 instances, each consisting of an egocentric utterance in Traditional Chinese, structured landmark coordinates, and the allocentric orientation label. Each instance includes a step-by-step reasoning trace for supervising CoT generation. To simulate speech-driven conditions, we use TTS and ASR to generate noisy transcripts. All instances are automatically produced from grid-based mapping rules and verified by human annotators.

**Data splits.** The dataset is divided into 3,216 training, 688 validation, and 696 test examples, with training using clean text and ASR transcripts only during evaluation. To avoid distributional bias, the training set is balanced to cover all combinations of spatial relations. Each single orientation

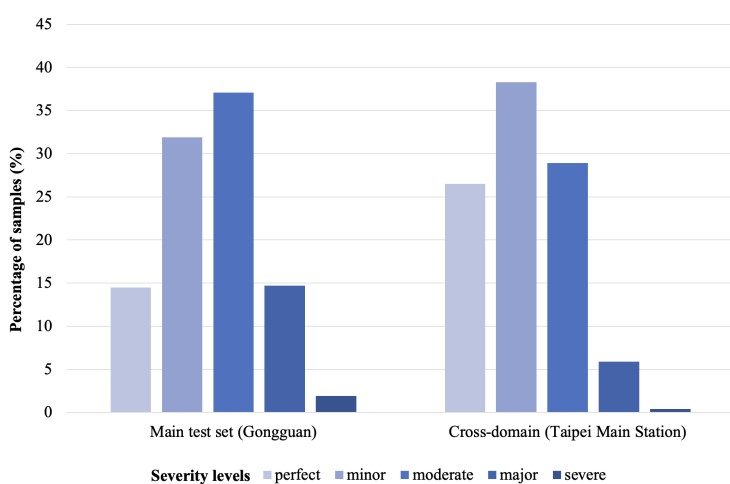

Figure 4: ASR error severity distribution in the two evaluation sets.

Table 2: Effectiveness on the egocentric spatial orientation task. Our MCoT with curriculum learning outperforms unimodal and non-structured baselines under both clean and ASR conditions.

| ID | Method | Setting | Acc. (%) | Format err. (%) | Reasoning |
|----|--------|---------|----------|-----------------|-----------|
| B1 | Zero-shot | Clean | 25.0 | 5.2 | – |
| B2 | Few-shot (no CoT) | Clean | 25.9 | 4.7 | – |
| B3 | Few-shot (with CoT) | Clean | 21.1 | 39.2 | 0.534 |
| B4 | Fine-tuned (no CoT) | Clean | 12.8 | 50.4 | – |
| B5a | **MCoT + curriculum (ours)** | **Clean** | **100.0** | **0.0** | **1.000** |
| B5b | **MCoT + curriculum (ours)** | **ASR** | **98.1** | **0.0** | **1.000** |

Reasoning quality reported only for methods with step-by-step reasoning; "–" means not applicable.

(front/back/left/right) contains 320 utterances. Double-orientation combinations (e.g., front+left, back+right, etc.), triple- and quadruple-orientation combinations, each contain about 280 utterances. This balanced design ensures fair coverage for training and evaluation. A subset of 400 examples introduces controlled linguistic variations (e.g., synonym substitutions, word-order changes), distributed across the splits. Test sets include multilingual elements common in Taiwan, with English landmark names appearing in 4.7% of main test samples and 46.5% of cross-domain samples, as well as occasional simplified Chinese variants from ASR outputs.

**Evaluation subsets.** Beyond the main test set (696 examples from Gongguan area), we build two additional evaluation sets for RQ3: (1) Cross-domain, 540 examples projected into a $10 \times 10$ grid from unseen Taipei Station area (Figure 5); and (2) Referential ambiguity, 200 cases with ambiguous references, disfluent or incomplete utterances, and semantically underspecified mentions.

**ASR error profile.** Figure 4 shows the ASR error severity distribution across the main and cross-domain test sets. Exact counts are provided in Appendix 5 (Table 7).

**Example.** *"I am at Gongguan MRT Exit 2, and restaurant 5 is on my right"* $\rightarrow$ Label: North.

## 4.2 EXPERIMENTS ON ORIENTATION REASONING (RQ1: METHOD EFFECTIVENESS)

We report *orientation accuracy* as the primary metric, the proportion of predictions matching the ground-truth orientation. For step-by-step methods, we also measure *reasoning quality*, the match rate of intermediate steps (0–1, higher is better), and *format error rate*, the proportion of outputs violating the expected schema. Table 2 summarizes results across baselines and our method.

Figure 5: Cross-domain evaluation environment. Taipei Station area projected into a $10 \times 10$ grid.

Figure 6: Representative error cases falling into three categories: direction understanding errors, relation extraction errors, and ASR misrecognition errors.

**Findings and error analysis.** Our curriculum-trained MCoT improves accuracy by 74.1 percentage points over the strongest baseline (B2), achieving 100.0% on clean inputs and 98.1% with ASR transcripts. In the ASR setting, the model makes 13 residual errors (13/696, 1.9%). These errors mostly involve direction understanding (9 cases), with relation extraction mistakes (2 cases) and ASR misrecognition errors (3 cases; categories may overlap). Representative examples are shown in Figure 6. Full reasoning traces for all residual cases are provided in Appendix A.1.

## 4.3 ABLATION STUDIES (RQ2: COMPONENT ANALYSIS)

We conduct ablation studies to assess the contributions of ASR transcripts, spatial coordinates, and structured CoT reasoning. Results are reported in Table 3.

**Findings.** Adding coordinates to ASR transcripts (A2→A3) improves accuracy by 10.2 percentage points and reduces format errors. Introducing structured CoT reasoning on top of multimodal

Table 3: Ablation study results. Spatial coordinates reduce ASR errors, while structured CoT provides the largest gains. The complete system achieves highest accuracy with no format errors.

| ID | Configuration | Accuracy (%) | Format error (%) |
|----|---------------|--------------|------------------|
| A1 | Clean text only (no coords) | 25.0 | 0.7 |
| A2 | ASR text only (no coords) | 16.2 | 35.8 |
| A3 | ASR text + coordinates (no CoT) | 26.4 | 3.0 |
| **A4a** | **Complete (clean text + coords + CoT)** | **100.0** | **0.0** |
| **A4b** | **Complete (ASR text + coords + CoT)** | **98.1** | **0.0** |

Table 4: Robustness and generalization results. The model maintains high accuracy across linguistic variations, unseen domains, and referential ambiguity.

| ID | Setting | Accuracy | Format error | Reasoning |
|----|---------|----------|--------------|-----------|
| R1 | Linguistic variation | 100% | 0.0% | 1.000 |
| R2 | Cross-domain (Taipei Station grid) | 94.6% (511/540) | 0.0% | 1.000 |
| R3 | Referential ambiguity | 99.5% (199/200) | 0.0% | 1.000 |

inputs (A3→A4b) contributes over 70 additional points. The complete system (A4a, A4b) eliminates all format errors, with 100% accuracy on clean inputs and 98.1% under ASR transcripts.

### 4.4 ROBUSTNESS ANALYSIS (RQ3: ROBUSTNESS AND GENERALIZATION)

We evaluate robustness to linguistic variation, cross-domain generalization, and referential ambiguity. Results are reported in Table 4. Detailed experimental configurations for each robustness test are provided in Appendix A.4.

**Findings and discussion.** Across R1–R3, the model sustains perfect reasoning quality and zero format errors. These results suggest that the structured three-step MCoT effectively handles linguistic variation, transfers to unseen domains (Taipei Station), and manages referential ambiguity. The 29 residual errors in R2 are mostly direction understanding errors (22/29), followed by ASR misrecognition errors (6/29) and relation extraction errors (1/29).

## 5 CONCLUSION AND LIMITATIONS

**Conclusion.** This paper introduced a MCoT framework for egocentric-to-allocentric orientation reasoning in conversational navigation. The three-step reasoning process achieves 100% accuracy on clean text and 98.1% on ASR transcripts in Traditional Chinese. Experiments show that (1) curriculum-trained MCoT substantially outperforms unimodal and non-structured baselines, (2) spatial coordinates and structured reasoning help the model remain accurate under noisy transcripts, and (3) the approach demonstrates generalization across linguistic variation, unseen domains, and referential ambiguity. Overall, the results indicate that structured reasoning can enhance both accuracy and interpretability in conversational navigation.

**Limitations and future work.** Despite these contributions, this study has limitations. Evaluation is conducted within a 10×10 grid environment, and the framework is developed primarily for Traditional Chinese using synthesized speech data with grid-based rules, human verification, and controlled variation to approximate realistic conversational navigation. Future work should extend to larger continuous environments, incorporate real-time multilingual speech recognition under diverse noise, and explore modalities such as vision or motion cues for more realistic settings.

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

# A APPENDIX

## A.1 ERROR TAXONOMY AND EXAMPLES

We categorize residual errors into three main types. Table 5 summarizes their definitions, and we provide one representative example for each type below.

| Error type | Definition and example |
|---|---|
| Direction Understanding | The model correctly extracts landmarks and computes their coordinates, but misapplies the spatial mapping rules between relative orientation terms (front, back, left, right) and absolute directions (North, South, East, West). |
| Relation Extraction | The model fails at relation extraction, misidentifying or omitting a landmark. Even if coordinates are computed correctly, reasoning is built on faulty relations. |
| ASR Misrecognition | Errors caused by transcription mistakes in the ASR system, which corrupt the input before reasoning. |

Table 5: Error taxonomy: three types of residual errors observed in our analysis.

**Direction Understanding Error**

**Model Input**

Audio: *"我現在在在台北車站S2出口，前面是餐廳5，左邊是台北車站S3出口"*
Coordinates: 台北車站S2出口(4,4), 餐廳5(4,5), 台北車站S3出口(3,4)
問題: 使用者面朝哪個方向？請逐步推理：

**Model Output**

第一步：提取空間關係
關係1：空間關係=前面，參考地標=餐廳5
關係2：空間關係=左邊，參考地標=台北車站S3出口
第二步：計算絕對方向
參考地標1=餐廳5，從台北車站S2出口到餐廳5方向向量：(4,5)-(4,4)=(0,1)，方向：北方
參考地標2=台北車站S3出口，從台北車站S2出口到台北車站S3出口方向向量：(3,4)-(4,4)=(-1,0)，方向：西方
第三步：推理朝向
餐廳5在北方且在使用者的左邊，空間映射：面朝南方時，左邊=北方
台北車站S3出口在西方且在使用者的左邊，空間映射：面朝南方時，左邊=西方
結論：使用者面朝南方

Final answer: 南方 ✗

Figure 7: Direction understanding error with original Chinese model output.

A.1.1 DIRECTION UNDERSTANDING ERROR

**Error Analysis.** The model correctly extracted relations and computed absolute directions but incorrectly applied spatial mapping rules in the final step, concluding the user faces South. Figure 7 shows the original output.

**Input.**

```
Audio:  I am at Taipei Main Station Exit S2, restaurant 5
is in front of me, and Taipei Main Station Exit S3 is on
my left | Coordinates:  Taipei_Main_Station_Exit_S2(4,4),
restaurant_5(4,5), Taipei_Main_Station_Exit_S3(3,4)
```

**Model Output.**

1. **Step 1: Extract spatial relations**
   Spatial relation 1 = front, landmark = restaurant 5
   Spatial relation 2 = left, landmark = Taipei Main Station Exit S3

2. **Step 2: Calculate absolute directions**
   Reference landmark 1 = restaurant 5, Direction vector from Taipei Main Station Exit S2 to restaurant 5: (4,5)-(4,4) = (0,1), Direction = North
   Reference landmark 2 = Taipei Main Station Exit S3, Direction vector from Taipei Main Station Exit S2 to Taipei Main Station Exit S3: (3,4)-(4,4) = (-1,0), Direction = West

3. **Step 3: Infer user orientation**
   Restaurant 5 is to the North and the user describes it as "on my left". Spatial mapping rules

Figure 8: Relation extraction error with original Chinese model output.

indicate left = North when the user is facing South.

Taipei Main Station Exit S3 is to the West and the user describes it as "on my left". Spatial mapping rules indicate left = West when the user is facing South.

**Final Answer.** The user is facing South.

A.1.2 RELATION EXTRACTION ERROR

**Error Analysis.** The model incorrectly extracted spatial relations from the input, misidentifying which landmark corresponds to which direction. Figure 8 shows the original output.

**Input.**

```
Audio:  I am at bus stop 2, Taipei Main Station Exit S3
is in front of me, Taipei Main Station Exit K7 is on my
left, and sports store 1 is on my right | Coordinates:
bus_stop_2(3,3), Taipei_Main_Station_Exit_S3(3,4),
Taipei_Main_Station_Exit_K7(2,3), sports_store_1(4,3)
```

**Model Output.**

1. **Step 1: Extract spatial relations**
   Spatial relation 1 = left, landmark = Taipei Main Station Exit S3
   Spatial relation 2 = right, landmark = sports store 1

2. **Step 2: Calculate absolute directions**
   Reference landmark 1 = Taipei Main Station Exit S3, Direction vector from bus stop 2 to Taipei Main Station Exit S3: (3,4)-(3,3) = (0,1), Direction = North
   Reference landmark 2 = sports store 1, Direction vector from bus stop 2 to sports store 1: (4,3)-(3,3) = (1,0), Direction = East

3. **Step 3: Infer user orientation**
   Taipei Main Station Exit S3 is to the North and the user describes it as "on my left". Spatial mapping rules indicate left = North when the user is facing South.
   Sports store 1 is to the East and the user describes it as "on my right". Spatial mapping rules indicate right = East when the user is facing South.

**Final Answer.** The user is facing South.

Figure 9: ASR misrecognition error with original Chinese model output.

### A.1.3 ASR Misrecognition Error

**Error Analysis.** ASR misrecognition caused the model to incorrectly map multiple spatial relations to the same landmark, leading to inconsistent reasoning. Figure 9 shows the original output.

**Input.**

```
Audio:  I am at park 4, yin-liu-dian 4 is in front of me,
981 is behind me, nian-bao-dian 3 is on my left, and 982 is
on my right | Coordinates:  park_4(8,5), drink_shop_4(7,5),
bar_1(9,5), bakery_3(8,4), bar_2(8,6)
```

**Model Output.**

1. **Step 1: Extract spatial relations**
   Spatial relation 1 = front, reference landmark = yin-liu-dian 4
   Spatial relation 2 = back, reference landmark = yin-liu-dian 4
   Spatial relation 3 = left, reference landmark = nian-bao-dian 3
   Spatial relation 4 = right, reference landmark = bar 2

2. **Step 2: Calculate absolute directions**
   Reference landmark 1 = yin-liu-dian 4, Direction vector from park 4 to yin-liu-dian 4: $(8,5)-(7,5) = (1,0)$, Direction = East
   Reference landmark 2 = drink shop 4, Direction vector from park 4 to drink shop 4: $(7,5)-(8,5) = (-1,0)$, Direction = West
   Reference landmark 3 = nian-bao-dian 3, Direction vector from park 4 to nian-bao-dian 3: $(8,4)-(8,5) = (0,-1)$, Direction = South
   Reference landmark 4 = bar 2, Direction vector from park 4 to bar 2: $(8,6)-(8,5) = (0,1)$, Direction = North

3. **Step 3: Infer user orientation**
   Yin-liu-dian 4 is to the East and the user describes it as "in front of me". Spatial mapping rules indicate front = East when the user is facing East.
   Drink shop 4 is to the West and the user describes it as "behind me". Spatial mapping rules indicate back = West when the user is facing East.
   Nian-bao-dian 3 is to the South and the user describes it as "on my left". Spatial mapping rules indicate left = South when the user is facing East.

Bar 2 is to the North and the user describes it as "on my right". Spatial mapping rules indicate right = North when the user is facing East.

**Final Answer.** East

## A.2 IMPLEMENTATION DETAILS FOR THE EXPERIMENTS

**Model Architecture.** We use Taiwan-LLM-13B-v2.0-Chat as the base model with LoRA fine-tuning under 4-bit quantization (rank $r = 16$, $\alpha = 32$, dropout rate 0.05).

**Training Parameters.** All experiments run for 5 epochs with batch size 1 and gradient accumulation steps of 32. The learning rate is set to $5 \times 10^{-5}$, with a maximum sequence length of 768 tokens. Training is conducted in bfloat16 mixed precision on NVIDIA A100 GPUs.

**Data Format.** Inputs follow the LLaMA chat template, combining ASR transcripts and spatial coordinates. The model is trained to output structured three-step reasoning traces aligned with the MCoT design.

**ASR Preprocessing.** Speech inputs were transcribed with Whisper-base (zh) using greedy decoding (temperature 0, no beam search) on 16 kHz mono audio. We report character error rate (CER) on the TTS→ASR loop as a proxy of noise, rather than Whisper's internal probabilities. TTS was used solely to synthesize controlled variants for evaluation experiments.

Table 6: Training and ASR preprocessing details.

| | |
|---|---|
| Model | Taiwan-LLM-13B-v2.0-Chat (LoRA, 4-bit) |
| Rank / $\alpha$ / Dropout | 16 / 32 / 0.05 |
| Optimizer | AdamW, lr $= 5 \times 10^{-5}$ |
| Batch size / Accumulation steps | 1 / 32 |
| Precision | bfloat16 |
| Hardware | NVIDIA A100 |
| ASR | Whisper-base (zh), greedy, 16 kHz mono |
| Metric | CER on TTS→ASR loop |

## A.3 FULL PROMPTS

### A.3.1 TRAINING PROMPTS

**Step 0: Spatial Relation Extraction**

**Task.** Extract all spatial relations and reference landmarks from the following sentence.

**Input.** "I am at Gongguan MRT Exit 1, Restaurant 4 is in front of me, Gongguan MRT Exit 2 is behind me, Bus Stop 1 is on my left, and Bike Rental Station 3 is on my right."

**Output.**

```
Spatial relation 1 = front, Reference landmark 1 =
Restaurant 4
Spatial relation 2 = behind, Reference landmark 2 = Gongguan
MRT Exit 2
Spatial relation 3 = left, Reference landmark 3 = Bus Stop 1
Spatial relation 4 = right, Reference landmark 4 = Bike
Rental Station 3
```

**Step 1: Vector Direction Learning**

**Input.**

```
From = (0,0), To = (1,0)
```

**Output.**

```
Direction vector:  (1,0) - (0,0) = (1,0), Direction = East
```

**Step 2: Orientation Inference**

**Task.** Infer user orientation based on the landmark absolute direction and spatial relation.

**Input.**

```
Landmark absolute direction = East
Spatial relation = behind
```

**Output.**

```
Analysis:  The landmark is to the East, and the user
describes it as ''behind me.''  Spatial mapping rules
indicate behind = East when the user is facing West.
Therefore, the user is facing West.
```

**Step 3: Complete Chain-of-Thought Reasoning**

**Input.**

```
Audio:  I am at Academic Building A, Student Activity Center
1 is in front of me, Academic Building B is behind me,
Parking Lot 2 is on my left, and Small Plaza 2 is on my
right | Coordinates:  Academic Building A(7,1), Student
Activity Center 1(6,1), Academic Building B(8,1), Parking
Lot 2(7,0), Small Plaza 2(7,2)
```

**Output.**

1. **Step 1: Extract spatial relations**
   Spatial relation 1 = front, Reference landmark 1 = Student Activity Center 1
   Spatial relation 2 = behind, Reference landmark 2 = Academic Building B
   Spatial relation 3 = left, Reference landmark 3 = Parking Lot 2
   Spatial relation 4 = right, Reference landmark 4 = Small Plaza 2

2. **Step 2: Calculate absolute directions**
   Reference landmark 1 = Student Activity Center 1, Direction vector: (6,1) - (7,1) = (-1,0), Direction = West
   Reference landmark 2 = Academic Building B, Direction vector: (8,1) - (7,1) = (1,0), Direction = East
   Reference landmark 3 = Parking Lot 2, Direction vector: (7,0) - (7,1) = (0,-1), Direction = South
   Reference landmark 4 = Small Plaza 2, Direction vector: (7,2) - (7,1) = (0,1), Direction = North

3. **Step 3: Infer user orientation**
   Student Activity Center 1 is to the West, and the user describes it as "in front of me." Spatial mapping rules indicate front = West when the user is facing West.
   Academic Building B is to the East, and the user describes it as "behind me." Spatial mapping rules indicate behind = East when the user is facing West.
   Parking Lot 2 is to the South, and the user describes it as "on my left." Spatial mapping rules indicate left = South when the user is facing West.
   Small Plaza 2 is to the North, and the user describes it as "on my right." Spatial mapping rules indicate right = North when the user is facing West.
   Therefore, the user is facing West.

A.3.2 BASELINE EVALUATION PROMPTS

**Training data format.** All training uses the LLaMA chat template with Traditional Chinese inputs and outputs, as Taiwan-LLM-13B-v2.0-Chat is optimized for Traditional Chinese. All training and

evaluation were conducted in Traditional Chinese. The prompts below are English translations for clarity and reproducibility.

**B1: Zero-shot baseline**

```
Question:  Audio:  I am at Gongguan MRT Exit 3,
and Dormitory 2 is on my right | Coordinates:
Gongguan_MRT_Exit_3(5,8), Dormitory_2(6,8)
Which direction is the user facing?  Please answer North,
South, East, or West.
Answer:
```

**B2: Few-shot prompting (no CoT)**

```
Instruction:  Based on the audio description and coordinate
information, determine which direction the user is facing.

Example:  Audio:  I am at the gym, and the pharmacy is in
front of me | Coordinates:  gym(4,6), pharmacy(4,7)
Answer:  North

Example:  Audio:  I am at the park, and the water park is
behind me | Coordinates:  park(0,0), water_park(0,1)
Answer:  South

Example:  Audio:  I am at the foundation, and the high
school is on my right | Coordinates:  foundation(0,7),
high_school(0,6)
Answer:  East

Example:  Audio:  I am at the cooperative store,
and the theater is on my left | Coordinates:
cooperative_store(8,8), theater(8,7)
Answer:  West

Question:  {user_input}
Answer:
```

**B3: Few-shot prompting with CoT**

```
Instruction:  Use three-step reasoning to determine the
user's facing direction given the audio description and
coordinates.
```

**Example 1**
```
Input: Audio:  I am at the gym, and the pharmacy is in
front of me | Coordinates:  gym(4,6), pharmacy(4,7)
Output:
Step 1:  Extract spatial relations
Spatial relation = front
Reference landmark = pharmacy
Step 2:  Calculate absolute directions
Direction vector from gym to pharmacy:  (4,7) - (4,6) =
(0,1)
Direction = North
Step 3:  Infer user orientation
The pharmacy is to the North, and the user describes it as
``in front of me.''
Spatial mapping rules indicate front = North when the user
is facing North.
Therefore, the user is facing North.
```

**Example 2**
```
Input: Audio:  I am at the park, and the water park is
```

```
behind me | Coordinates:  park(0,0), water_park(0,1)
Output:
Step 1:  Extract spatial relations
Spatial relation = behind
Reference landmark = water park
Step 2:  Calculate absolute directions
Direction vector from park to water park:  (0,1) - (0,0) =
(0,1)
Direction = North
Step 3:  Infer user orientation
The water park is to the North, and the user describes it as
``behind me.''
Spatial mapping rules indicate behind = North when the user
is facing South.
Therefore, the user is facing South.

Now use the same three-step reasoning:
Input:  {user_input}
Output:
```

**B4: Fine-tuned direct classification**

```
USER: {user_input}
ASSISTANT:
```

## A.4 SPATIAL ROBUSTNESS DETAILS

### A.4.1 LINGUISTIC VARIATION ROBUSTNESS (R1)

To evaluate robustness to natural linguistic variations, we constructed test sets in Traditional Chinese with diverse expression patterns, while ensuring identical spatial semantics and orientation outputs.

**Variation types**

- **Word order variations:** sentence inversion, argument permutation, and syntactic paraphrasing.
- **Synonym substitutions:** spatial term substitution, position verb substitution, and landmark term substitution.

### A.4.2 REFERENTIAL AMBIGUITY ROBUSTNESS (R3)

We test the model's ability to handle ambiguous or underspecified references commonly encountered in natural conversational navigation.

**Variation types**

- **Referential ambiguity:** generic references ("this building"), and demonstrative pronouns ("that place").
- **Incomplete utterances:** disfluency ("I am at... um..."), uncertainty markers ("should be"), and hesitation patterns.
- **Semantic underspecification:** vague location terms ("some building") and imprecise references ("over there").

**Example of R3 test cases**

```
Original:  Audio:  I am at security office, and Dormitory
6 is behind me | Coordinates:  security_office(7,3),
dormitory_6(7,4)
Referential ambiguity:  Audio:  I am at this building, that
dormitory is behind me | Coordinates:  security_office(7,3),
dormitory_6(7,4)
```

**Incomplete utterance:** `Audio: I am at... um... security office, Dormitory 6 should be behind | Coordinates: security_office(3,5), dormitory_6(3,4)`

**Semantic underspecification:** `Audio: I am at some place, that building over there is behind me | Coordinates: security_office(3,5), dormitory_6(3,4)`

## A.5 ASR ERROR SEVERITY STATISTICS

Table 7 reports the distribution of ASR error severity in both evaluation sets.

Table 7: ASR error severity distribution with exact counts.

| Evaluation set | Severity | Count | Percent |
|---|---|---|---|
| Main test set (Gongguan) | perfect | 101 | 14.5% |
| | minor | 222 | 31.9% |
| | moderate | 258 | 37.1% |
| | major | 102 | 14.7% |
| | severe | 13 | 1.9% |
| Cross-domain (Taipei Station) | perfect | 143 | 26.5% |
| | minor | 207 | 38.3% |
| | moderate | 156 | 28.9% |
| | major | 32 | 5.9% |
| | severe | 2 | 0.4% |

