# OpenReview forum: "Conversational Orientation Reasoning: Egocentric-to-Allocentric Navigation with Multimodal Chain-of-Thought"
_ICLR.cc/2026/Conference — ICLR 2026 Conference Withdrawn Submission_

### Official Review · Reviewer_Kvna · 2025-10-25

**Soundness:** 1
**Presentation:** 3
**Contribution:** 2
**Rating:** 2
**Confidence:** 1

**Summary:**

This paper proposed a Conversational Orientation Reasoning (COR) benchmark for Traditional Chinese conversational navigation. To solve this languages based, tabular egocentric-to-allocentric navigation task, authors further proposed a multi-modal chain-of-thought (MCoT) framework for fine-tuning a Taiwan-LLM-13B-v2.0-Chat. The proposed framework can solve the egocentric-to-allocentric with high success rate and showed robustness to noise in the conversation input.

Nevertheless, the benchmark itself consisted a simple 10x10 table, which could oversimplify the realistic task. One can imagine that the realistic task could involve high-dimensional input, e.g., images captured from the user or vectorized map. A multi-model LLM could have the capability to solve this high-dim to orientation mapping task, rather than tabular setup.

**Strengths:**

The paper proposed a novel problem and a solution on conversational orientation reasoning from language and semantic map. Experiments demonstrated the effectiveness and robustness of the proposed solution. The paper presented the problem and the method clearly.

**Weaknesses:**

The major weakness is the proposed problem oversimplified real-world scenarios. This surprised the capability of the method as well as multi-model LLM’s capabilities. The current method seems can only solve an in-distribution 10x10 table and another novel 10x10 table. Reasoning the orientation from a table seems can be solve using hard-code.

**Questions:**

What is the advantage of the proposed method over a set of hard-coded rules?
Could proposed method reason from high-dimensional input, for example, from the photos captured from a user?
How capable is the method generalize to novel tables or even open-world language/landmarks?

---

### Official Review · Reviewer_zCUP · 2025-10-30

**Soundness:** 2
**Presentation:** 3
**Contribution:** 1
**Rating:** 2
**Confidence:** 3

**Summary:**

- The authors propose a benchmark for conversational navigation that is in Chinese, a multimodal CoT framework, and a curriculum learning strategy to try and achieve high performance on this benchmark. Overall, the authors are able to achieve strong results on the proposed benchmark by fine-tuning an open LLM using their proposed framework.

**Strengths:**

- The authors achieved strong performances on the proposed benchmark.
- The figures are clear.

**Weaknesses:**

- Nothing proposed in the paper is unique or *novel*, everything done was a standard fine-tuning technique with an added curriculum and standard multimodal reasoning steps.
- The central research questions are not particularly exciting, important, or impactful for the field
- The model achieves 100% reasoning performance on on the proposed egocentric spatial orientation task, which signals two things to me:
	- First, the model is likely overfitting this task after being fine-tuned, resulting in a loss of much of its prior knowledge.
	- Second, the proposed benchmark is too easy.
- While some of these ideas are briefly mentioned in the limitations, I see them as notable large problems.
	- The models are evaluated in an unrealistic environment (a 10x10) grid.
	- The authors only focus on a single language.
- The proposed approach is complex, and not easy to implement.

**Questions:**

N/A

---

### Official Review · Reviewer_dPzU · 2025-11-01

**Soundness:** 2
**Presentation:** 1
**Contribution:** 2
**Rating:** 2
**Confidence:** 4

**Summary:**

This paper's topic is translating egocentric utterances (e.g., on my right) into allocentric (N/E/S/W) orientation in conversational navigation. It introduces COR, a Traditional-Chinese benchmark drawn from real urban layouts projected onto a 10×10 Manhattan grid, where inputs are ASR-transcribed speech plus landmark coordinates and outputs are cardinal orientations. The authors propose a multimodal chain-of-thought (MCoT) framework with a structured three-step reasoning recipe: (1) relation extraction, (2) coordinate to absolute direction mapping, (3) final orientation inference, which is trained via curriculum on Taiwan-LLM-13B-v2.0-Chat. On COR, MCoT reports 100% accuracy on clean transcripts and 98.1% with ASR transcripts, with additional robustness under linguistic variation, cross-domain transfer (Taipei Station), and referential ambiguity. The paper argues that structured reasoning improves both accuracy and interpretability for resource-constrained, GPS-limited scenarios.

**Strengths:**

1. The task is formalized cleanly with explicit mapping rules and a Manhattan grid, focused on the neglected egocentric to allocentric transformation rather than high-level action prediction, and does so in a non-English, ASR-noisy setting (Traditional Chinese). The three-step MCoT plus curriculum addresses reasoning stability and interpretability.
2. The method addresses GPS-denied environments and resource-constrained deployments. The reported accuracy suggests potential utility for speech-driven navigation assistants where full maps/sensors are unavailable.

**Weaknesses:**

1. The task on a 10 x 10 Manhattan grid with axis-aligned landmarks and a fixed rule table (Table 1) looks algorithmically solvable by a simple deterministic program: parse the relation and landmark, compute $\Delta$, take `AbsDir(Δ)` by comparing $|\Delta_x|$ vs $|\Delta_y|$, then rotate by the relation. Without a strong non-neural baseline, the 100% clean accuracy is hard to contextualize.
I suggest adding (i) a rule-based solver and (ii) a probabilistic variant robust to noisy extraction, and comparing accuracy and latency.
2. The ASR noise is introduced via a TTS to ASR loop on clean, templated text, not real user speech. CER-based severity is reported, but real conversational prosody, disfluency, and OOVs can be harsher.
3. The grid is small, axis-aligned, and excludes diagonal cases; mapping rules are deterministic and known.
4. Baseline choices and numbers look unusually weak.
   Few-shot (with/without CoT) and fine-tuned no CoT baselines perform near chance or worse, which raises questions about prompt/format mismatches rather than inherent task difficulty.
5. Data are programmatically generated from the same mapping rules used by the model’s step-2/3 supervision. Curriculum-tuning on these deterministic recipes could lead to memorization of rule templates rather than robust reasoning.
6. Reasoning quality is reported as a match rate of intermediate steps and format error as schema violations, but neither measures *faithfulness* (whether the trace genuinely causes the answer).
7. The scope of robustness is still narrow.
   Cross-domain stays on a 10 x 10 Manhattan grid with the same rule table. Strong performance may reflect data homogeneity.
8. The work motivates resource-constrained, GPS-limited settings but evaluates offline on a mid-size 13B chat model with LoRA.
I suggest providing latency/memory profiles, and a small-footprint variant (e.g., 1–3B or distilled student) to substantiate edge deployment claims.

**Questions:**

1. What is the performance and latency of a deterministic solver that (a) extracts the relation via a regex/IE component and (b) applies Table 1 + rotation? If omitted, why?
2. How did you ensure the programmatic generation templates and the model’s step-wise supervision do not leak distributional shortcuts that trivialize the task?
3.  Is the reasoning trace *required* to be correct for the final answer to be accepted (e.g., by an external checker), or can the model produce a correct label with an incorrect intermediate step?
4. Beyond the TTS to ASR loop, do you have evaluations on spontaneous human speech (code-switching, disfluency, accents) recorded in the target locales?
6. How would MCoT handle diagonal/off-axis landmarks, continuous coordinates, or non-orthogonal street plans?
8. How sensitive is performance to removing supervision on *one* of the three steps (e.g., only supervise steps 1+3)? Any evidence that curriculum (vs. joint training) is the key driver?

---

### Note · Authors · 2025-11-14

I have read and agree with the venue's withdrawal policy on behalf of myself and my co-authors.